# Asymmetric Decision-Making in Online Knowledge Distillation: Unifying Consensus and Divergence

Zhaowei Chen [* 1]  Borui Zhao [* 1]  Yuchen Ge [2]  Yuhao Chen [1]  Renjie Song [1]  Jiajun Liang [1]

## Abstract

Online Knowledge Distillation (OKD) methods streamline the distillation training process into a single stage, eliminating the need for knowledge transfer from a pretrained teacher network to a more compact student network. This paper presents an innovative approach to leverage intermediate spatial representations. Our analysis of the intermediate features from both teacher and student models reveals two pivotal insights: (1) the similar features between students and teachers are predominantly focused on foreground objects. (2) teacher models emphasize foreground objects more than students. Building on these findings, we propose Asymmetric Decision-Making (ADM) to enhance feature consensus learning for student models while continuously promoting feature diversity in teacher models. Specifically, Consensus Learning for student models prioritizes spatial features with high consensus relative to teacher models. Conversely, Divergence Learning for teacher models highlights spatial features with lower similarity compared to student models, indicating superior performance by teacher models in these regions. Consequently, ADM facilitates the student models to catch up with the feature learning process of the teacher models. Extensive experiments demonstrate that ADM consistently surpasses existing OKD methods across various online knowledge distillation settings and also achieves superior results when applied to offline knowledge distillation, semantic segmentation and diffusion distillation tasks.

*Equal contribution [1]JIIOV Technology [2]University of Southern California. Correspondence to: Zhaowei Chen <zhaowei.chen@jiiov.com>, Jiajun Liang <tracyliang18@gmail.com>.

*Proceedings of the 42nd International Conference on Machine Learning*, Vancouver, Canada. PMLR 267, 2025. Copyright 2025 by the author(s).

## 1. Introduction

In conventional offline knowledge distillation paradigms (Hinton et al., 2015; Tian et al., 2019; Chen et al., 2021), a pre-trained teacher model provides soft supervision to guide student training through post-hoc knowledge transfer. This two-stage paradigm inherently suffers from computational redundancy and temporal decoupling between teacher-student interactions. In contrast, recently proposed Online Knowledge Distillation (OKD) methods simplify to one-stage distillation where both teacher and student models learn simultaneously from scratch, while achieving competitive or even superior performance (Guo et al., 2020; Qian et al., 2022; Song et al., 2023; Zhang et al., 2023; 2018b). Typically, Deep Mutual Learning (DML) (Zhang et al., 2018b) encourages each network to mutually learn from each other by matching their output logit via KL divergence. In this paper, we propose a novel Asymmetric Decision-Making strategy that unifies consensus and divergence feature learning to achieve better OKD performance.

This paper investigates a fundamental question: How does intermediate feature alignment affect model behavior in online distillation? Through systematic analysis using Class Activation Mapping(CAM) (Zhou et al., 2016) (Figure 1 and 2), we reveal two pivotal findings: (1) **the similar features between student and teacher models are predominantly concentrated on foreground objects**, suggesting students prioritize learning these "easy" patterns. (2) **teacher models highlight foreground objects more than students**, indicating an underutilized capacity for discriminative feature discovery. Surprisingly, the most informative divergent features (which may explain teacher superiority) still concentrate on foreground areas rather than background contexts.

Building on these insights, we propose Asymmetric Decision-Making (ADM). It is a unified framework that coordinates consensus enhancement and divergence exploration through spatially-aware feature modulation. Unlike conventional symmetric distillation, ADM implements role-specific learning strategies: For students, we amplify feature attribution in teacher-aligned foreground regions to accelerate confidence building. For teachers, we intensify exploration of under-activated foreground patterns to avoid pre-

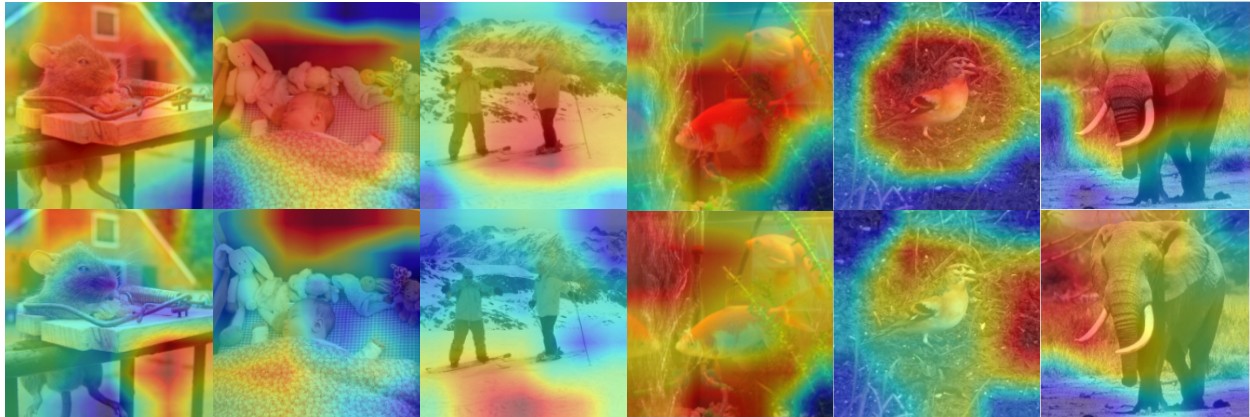

Figure 1: **Observation of Discrepancy Regions between Teacher and Student Features.** The first row shows that discrepancy regions between teacher and student's features are more concentrated on foreground object regions after Vanilla training. In contrast, the second row displays discrepancy regions are mainly found in background regions after ADM training. Best viewed in color.

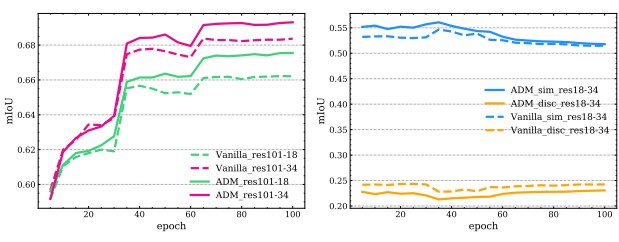

(a) mIoU of CAM regions.

(b) mIoU of similar and discrepancy regions.

Figure 2: **Analysis of the Teacher and Student Models' Intermediate Features.** We load a pre-trained ResNet101 model and obtain its CAM as the Target Object Regions (dubbed TOR) on ImageNet. (a) **mIoU of CAM regions.** We calculate the mean intersection over union (mIoU) between TOR and the CAM of intermediate ResNet-34, ResNet-18 models saved during the training process. We observe that ResNet-34 is more concentrated on object regions than ResNet-18. (b) **mIoU of similar and discrepancy feature regions.** We observe that the similar regions between ResNet-34 and ResNet-18 predominantly focus on TOR and a substantial proportion of discrepancy regions are also located within the TOR.

mature convergence to student-level representations. This strategic asymmetry creates a dynamic where students progressively close the performance gap while teachers continuously unveil new discriminative features. Our main contributions can be summarised as follows.

- Through interpretability-driven analysis, we identify the paradoxical coexistence of foreground-centric feature consensus and teacher-student attribution gaps, challenging conventional assumptions about knowledge transfer in online settings.

- We propose a simple but effective learning strategy named ADM, which allocates different attention to feature maps based on spatial similarity. ADM encourages student models to boost the attribution scores on foreground objects while reinforce the strengths of teacher models continually.

- Comprehensive experiments are conducted to verify our effectiveness, including online KD, offline KD, semantic segmentation and diffusion distillation. It is shown that the proposed method consistently outperforms state-of-the-art OKD methods.

## 2. Related Work

**Knowledge Distillation** KD aims to transfer knowledge acquired by an effective but cumbersome teacher model to a smaller compact student model. KD has attracted wide interest in vision and language applications (Meng et al., 2022; Niu et al., 2022; Sanh et al., 2019). KD can be formulated in logits-based (Hinton et al., 2015; Zhao et al., 2022; Huang et al., 2022; Sun et al., 2024), feature-based (Heo et al., 2019; Chen et al., 2021; Li, 2022; Xiaolong et al., 2023; Huang et al., 2024), and relation-based (Tian et al., 2019). Recent studies have focused on automating the process of discovering effective knowledge distillation strategies or architectures (Li et al., 2024; 2023; Dong et al., 2023; Hao et al., 2024). Besides, for the foreground and background imbalance in object detection, FGD (Yang et al., 2022) proposed to use ground-truth boxes to separate the images and guide the student to focus on crucial pixels and channels through spatial and channel attention. GID (Dai et al., 2021) select the discriminative instances with higher scores for distillation in object detection. These two methods under comparison primarily rely on bounding boxes to explicitly discern the foreground and background for distil-

lation, which markedly contrasts with our proposed method.

**Online Knowledge Distillation** The traditional offline KD framework needs a pre-trained and fixed teacher model. Online KD methods are more attractive because the training process is simplified to a single stage (Li et al., 2022; 2021; Lin et al., 2022; Yang et al., 2023). Deep Mutual Learning (DML) (Zhang et al., 2018b) enables students to share knowledge from each other's predictions to achieve distillation without teachers. KDCL (Guo et al., 2020) integrates the output of multiple students under different data augmentations as soft labels to guide students to optimize. SwitOKD (Qian et al., 2022) attempts to close the accuracy gap by slowing down the training of the teacher during training. SHAKE (Li & Jin, 2022) bridges offline and online knowledge transfer by building an extra shadow head as a proxy teacher model to perform mutual distillation with the student model. Recently proposed online KD methods mainly focus on logits-based. In this work, ADM unifies consensus and divergence feature learning for teacher and student models, which leverages the spatial similarity between teacher and student models, thus implicitly distinguishing between the foreground and background. Moreover, the foreground and background in our method embody a crucial attribute, representing simple and hard regions respectively. Consequently, our approach facilitates the student model to learn from the teacher model progressively, following a gradient of difficulty that ranges from easy to challenging tasks. This constitutes an innovative perspective that has not previously been proposed in the existing literature.

**Diffusion Distillation** Diffusion distillation methodologies can be taxonomically categorized into step distillation and model distillation. (Salimans & Ho, 2022; Meng et al., 2023) established the iterative progressive distillation framework, compressing DDPM sampling steps from 1,000+ to 4 through cyclic teacher-student optimization. The Dream-Fusion paradigm (Poole et al., 2022) first bridged text-to-3D generation through differential rendering and SDS losses. Recent work (Nguyen & Tran, 2024; Yin et al., 2024) achieves 40× acceleration via adaptive step annealing. Adversatial Distillation (Sauer et al., 2025) integrated adversarial loss to ensure high image fidelity even in the low-step regime of one or two sampling steps. (Kim et al., 2025) proposed block pruning and feature distillation for low-cost general-purpose Text-to-image (T2I) generation. In this paper, we integrate ADM into both step distillation and model distillation to verify its effectiveness.

## 3. Preliminary

### 3.1. Online Knowledge Distillation

Generally, OKD replaces the commonly used pre-trained teachers with peer-trainable models. The training loss con-

sists of the Cross-Entropy (CE) loss and the Knowledge Distillation (KD) loss. Let $\mathcal{D} = \{x_i, y_i\}_{i=1}^N$ be a training set containing $N$ images and $C$ categories of labels. $y_i \in \{1, \cdots, C\}$ is the ground-truth label of the image $x_i$. The $m$-th model ($m \in \{1, \cdots, M\}$) obtains its output logits $\mathbf{z}^m = f(x; \theta^m) \in \mathcal{R}^C$, where $\theta^m$ is the training parameters of the model $m$. The classification loss is calculated by Cross-Entropy:

$$\mathcal{L}_{ce}(\mathbf{z}^m, y_i) = -\log \frac{\exp(\mathbf{z}_{y_i}^m)}{\sum_{c=1}^C \exp(\mathbf{z}_c^m)}, \qquad (1)$$

The KD loss usually minimizes the KL divergence between the probabilistic outputs of different models:

$$\mathcal{L}_{kd} = \tau^2 \mathcal{D}_{KL}(p^m, p^n) = \tau^2 \sum_{c=1}^C p_c^n \log \frac{p_c^n}{p_c^m}, \qquad (2)$$

where $p^m, p^n \in \mathcal{R}^C$ are the soft logits produced by models $m$ and $n$. The soft logits are calculated by:

$$p^m = \sigma(\mathbf{z}^m/\tau) = \frac{\exp(\mathbf{z}^m/\tau)}{\sum_c \exp(\mathbf{z}_c^m/\tau)}, \qquad (3)$$

where $\sigma$ is the softmax function, and $\tau$ is the temperature factor to control the softness of logits. Then the DML loss is formulated as follows and $\lambda$ is the trade-off weight:

$$\mathcal{L}_{dml} = \sum_{i=1}^M (\mathcal{L}_{ce}^i + \lambda \mathcal{L}_{kd}^i), \qquad (4)$$

### 3.2. Online Feature Distillation

In recent offline KD methods, FitNet(Romero et al., 2014) combines the feature-based loss with the logit-based loss to improve the performance. In online KD, we observed that making the teacher mimic the student's features would compromise the diversity of the teacher's features, thereby leading to detrimental effects on performance. Due to the accelerated convergence and higher accuracy of teacher models, teachers exhibit superior features that effectively facilitate the learning process for students. Thereby, we simply use the following MSE loss for the student feature distillation but detach the gradient of teacher feature distillation:

$$\mathcal{L}_{feat} = \frac{1}{B} \sum_{b=1}^B \frac{1}{HWC} \sum_{i=1}^H \sum_{j=1}^W \sum_{c=1}^C (F_{bijc}^t - \phi(F_{bijc}^s))^2, \qquad (5)$$

where $B$ is the batch size. $i, j$ are the location of corresponding feature map with width $W$ and height $H$. $C$ represents the number of channels. $F^t$ and $F^s$ are the intermediate features of student and teacher models, respectively. The function $\phi$ is the $1 \times 1$ convolutional adaptation function to adapt $F^s$ to the same dimension as $F^t$.

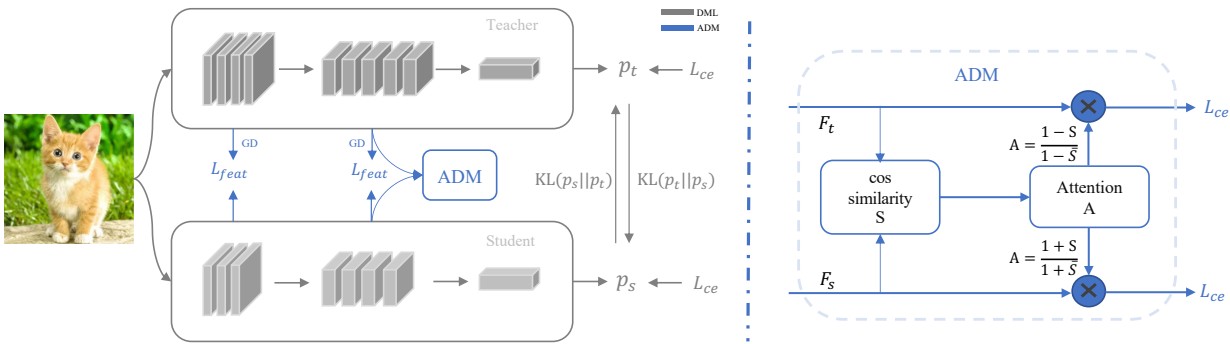

Figure 3: **Asymmetric Decision-Making in Online Knowledge Distillation.** GD means Gradient Detach operation. Consensus Learning for student models and Divergence Learning for teacher models. The gray arrow represents the baseline of the DML method, while the blue arrow denotes the newly added ADM component. Best viewed in color.

## 4. Method

### 4.1. Asymmetric Decision-Making in Online Knowledge Distillation

In this section, we show how to design Asymmetric Decision-Making in Online Knowledge Distillation. Intuitively, during the training, the teacher's foreground object-centric features should be strengthened while the student features should be changed accordingly. The above analysis suggests that ADM function can be decomposed into two components, namely, Consensus loss $\mathcal{L}_{co}$ and Divergence loss $\mathcal{L}_{di}$. $\mathcal{L}_{co}$ is applied to student model, indicating that the higher similarity between the student and teacher models in specific regions, the stronger the supervision we provide for those regions. The similarity matrix is described as:

$$\mathcal{S}_{ij} = \mathcal{D}(F_{ij}^t, F_{ij}^s) = \frac{F_{ij}^t F_{ij}^{s\,T}}{\left\| F_{ij}^t \right\|_2 \left\| F_{ij}^s \right\|_2}, \quad (6)$$

where $F^t, F^s \in \mathcal{R}^{h \times w \times d}$, $\mathcal{S} \in \mathcal{R}^{h \times w}$, $d$ is the channel number. $i$ and $j$ are the location of the corresponding feature map with width $w$ and height $h$. As shown in Figure 3. For simplicity, we only take the penultimate feature before Global Average Pooling (GAP) as input. $\mathcal{L}_{co}$ encourages the student to focus on learning simple knowledge rather than concentrating on difficult regions where a large discrepancy with the teacher model, described as:

$$\mathcal{L}_{co} = \mathcal{L}_{ce}\left(W^s\left(\psi_{avg}\left(\frac{1+\mathcal{S}}{1+\bar{\mathcal{S}}} \cdot F^s\right)\right), y_i\right), \quad (7)$$

where $W^s$ is the parameters of the last student linear classifier. $\psi_{avg}$ is GAP operation. $\bar{\mathcal{S}}$ is the mean value of $\mathcal{S}$ along the dimension of $h$ and $w$. The Attention Function for students is a monotone-increasing function. Simultaneously, to avoid the teacher model learning an excessively high similarity with the student model and falling into a local

optimum, we introduce $\mathcal{L}_{di}$ to the teacher model, allowing it to continuously strengthen the learning of discrepancy regions from the student model, described as:

$$\mathcal{L}_{di} = \mathcal{L}_{ce}\left(W^t\left(\psi_{avg}\left(\frac{1-\mathcal{S}}{1-\bar{\mathcal{S}}} \cdot F^t\right)\right), y_i\right), \quad (8)$$

where $W^t$ is the parameters of the last teacher linear classifier. The Attention Function for teachers is a monotone-decreasing function. By summing up both losses as follows, the teacher model constantly guides and drives the student model to learn and catch up. $\alpha$ and $\beta$ are hyper-parameters. We here employ CE loss function in ADM for conciseness, and in addition, as shown in Section 5.7, we have investigated the combination of CE loss with KD loss, which achieved competitive results.

$$\mathcal{L}_{adm} = \alpha\mathcal{L}_{co} + \beta\mathcal{L}_{di}, \quad (9)$$

As a result, the overall training loss $\mathcal{L}_{total}$ can be composed of DML loss, feature loss, and adm loss:

$$\mathcal{L}_{total} = \mathcal{L}_{dml} + \gamma\mathcal{L}_{feat} + \mathcal{L}_{adm}, \quad (10)$$

where $\gamma$ is a factor for balancing the losses. In this way, via the ADM loss, we have enabled the student to match the teacher network's feature adaptively, thus boosting OKD performance to a great extent.

## 5. Experiment

### 5.1. Experimental settings

**Datasets.** We validated ADM on the following datasets: **CIFAR-10** (Krizhevsky et al., 2009) consists of 60000 32x32 colour images in 10 classes, with 6000 images per class. **CIFAR-100** (Krizhevsky et al., 2009) consists of 32×32 color images drawn from 100 classes, which are split into 50k train and 10k test images. **ImageNet** (Rus-

sakovsky et al., 2015) is a large scale image classificaiton dataset that contains 1k classes with about 1.28 million training images and 50k images for validation. Each image is resized to 224x224. On ImageNet and CIFAR-100, we adopt the commonly used random crop and horizontal flip techniques for data augmentation. **Cityscapes** (Cordts et al., 2016) is a dataset for urban scene parsing, comprising 5000 meticulously annotated images. The distribution of these images is as follows: 2975 for training, 500 for validation, and 1525 for testing.

**Teacher-Student Pairs.** To validate the generalizability of different distillation methods, we select a group of popular network architectures to form different teacher-student pairs. By combining different teacher and student networks, like ResNets (He et al., 2016), WideResNet(WRN)s (Zagoruyko & Komodakis, 2016), VGGs (Simonyan & Zisserman, 2014), MobileNets (Howard et al., 2017; Sandler et al., 2018), and ShuffleNets (Ma et al., 2018; Zhang et al., 2018a), we can perform distillation between the same architectures (e.g., ResNet34 (He et al., 2016)-ResNet18(He et al., 2016)) and different architectures (e.g., ResNet50(He et al., 2016)-MobileNet-V2(Sandler et al., 2018)). ADM introduces no additional trainable parameters. The added computational cost is markedly minimal relative to the original. For example, in the ResNet34-18 experiments on ImageNet, ResNet34 has 3.67G FLOPs and ResNet18 has 1.82G FLOPs. ADM only adds 0.52M FLOPs.

**Implementation Details.** Following the settings of previous methods (Qian et al., 2022; Tian et al., 2019), the batch size, epochs, learning rate decay rate, and weight decay rate are 256/128, 100/300, 0.1/0.1, and 0.0001/0.0005, respectively on ImageNet/CIFAR-100. The initial learning rate is 0.1 on ImageNet, and 0.01 for MobileNetV2, 0.1 for the other students on CIFAR-100. Besides, the learning rate drops every 30 epochs on ImageNet and drops at 140, 200, 250 epochs on CIFAR-100. The optimizer is Stochastic Gradient Descent (SGD) with momentum 0.9. By following the conventions in OKD, we use a fixed temperature T as 1.0 and loss weight $\lambda$ as 1.0 for all experiments. For a fair comparison, the hyper-parameters of different methods are fixed in all experiments. We set $\alpha = 0.01$, $\beta = 0.01$, $\gamma = 1.0$ on CIFAR-100 dataset because of limited spatial information, and $\alpha = 0.2$, $\beta = 0.6$, $\gamma = 0.01$ on ImageNet dataset for all OKD experiments. It's worth noting that ADM exhibits no sensitivity to hyperparameter settings as shown in Table 10 and 12.

### 5.2. Online KD Experiments on Image Classification

**Results on CIFAR-100.** Table 1 and Table 2 report the experimental results on CIFAR-100 with similar architecture style and dissimilar architecture style teacher-student pairs, respectively. **We run all methods five times with differ-**

**ent seeds and obtain the average accuracy.** ADM is a plug-and-play module that can effectively enhance the performance of existing methods. Note that ADM can perform much better, especially for heterogeneous student-teacher pairs, +0.35% vs +1.41%. This is because heterogeneous architectures focus on different regions on intermediate features. It is more discriminative for students to leverage the feature similarity compared to output logits.

| Backbone | Vanilla | DML | KDCL | SwitOKD | DML+ADM | SwitOKD+ADM |
|---|---|---|---|---|---|---|
| ResNet8×4 | 72.33 | 73.95 | 74.61 | 74.49 | 74.23 | **74.63** |
| ResNet32×4 | 79.55 | 79.99 | 79.84 | 79.31 | 79.62 | **80.01** |
| VGG-8 | 70.58 | 72.45 | 71.78 | 72.71 | 72.8 | **72.94** |
| VGG-13 | 75.12 | 76.19 | 74.63 | 75.64 | 76.07 | **76.50** |

Table 1: **Top-1 Accuracy (%) comparison on CIFAR-100 with same architecture style teacher-student pairs.** The upper and lower models denote student and teacher, respectively.

| Backbone | Vanilla | DML | KDCL | SwitOKD | DML+ADM |
|---|---|---|---|---|---|
| 0.5MobileNetV2 | 60.39 | 66.69 | 65.98 | 66.68 | **68.10** |
| WRN-16-2 | 73.01 | 73.59 | 71.54 | 73.55 | **73.62** |
| ShuffleNetV1 | 68.44 | 73.58 | 73.91 | 73.82 | **74.61** |
| ResNet32x4 | 79.30 | **80.08** | 78.97 | 79.37 | 79.97 |
| ShuffleNetV2 | 70.17 | 75.15 | 75.00 | 75.18 | **75.75** |
| ResNet32×4 | 79.38 | 80.15 | 79.14 | 79.90 | **80.26** |

Table 2: **Top-1 Accuracy (%) comparison on CIFAR-100 with different architecture style teacher-student pairs.** The upper and lower models denote student and teacher, respectively.

**Results on ImageNet.** Table 3 reports the performance of our approach on ImageNet with similar ResNets architectures. We run all methods three times with different seeds and obtain the Top-1 average accuracy. As presented in the table, ADM outperforms other OKD methods in most cases. Equipped with ADM, the DML baseline of ResNet-18 (11.7M parameters) gains 0.47% in Top-1 accuracy. Besides, we observe that a larger teacher model ResNet-101 (44.5M parameters) fails to distill a better compact student in other OKD methods. But in the paradigm of ADM, due to the teacher's superior performance, more confidence can be set in the discrepancies regions between the teacher and student, which, in turn, students learn uncomplicated features rather than challenging ones. As a result, ADM can effectively bridge the gap between the large teacher and student. In addition, even for two identical models, ADM can still enhance both performances. As shown in Table 4, With pair ResNet18 and 0.5MobileNetV2 (1.97M parameters), ADM obtains 0.89% accuracy gains than DML, and ResNet50 (25.6M parameters)-MobileNetV1 (4.23M parameters) pair 0.43% gains, verifying its effectiveness on heterogeneous architectures. It can be seen that *the relative difference in parameter size is not the main reason why a larger model*

*with better accuracy cannot serve as a good teacher for smaller models.* How to choose a good teacher model for distillation remains an open question worth discussing.

It's worth mentioning that all existing OKD methods encountered an identical issue. The performance gain on the homogenous backbone pair is not obvious compared to the heterogenous pair. There are primarily two reasons for this observation. Firstly, heterogeneous pairs exhibit greater structural differences, enabling them to acquire a broader range of knowledge from one another. Secondly, and more significantly, the student model within heterogeneous pairs is smaller. The performance disparity between the student and teacher models is more pronounced, resulting in a larger potential for improvement. An additional experimental observation is that the degradation of the teacher model's performance is a common issue among all existing OKD methods. However, our method outperforms other baseline approaches in terms of the teacher's performance improvement.

| Backbone | Vanilla | DML | KDCL | SwitOKD | DML+ADM |
|---|---|---|---|---|---|
| ResNet-18 | 70.16 | 71.12 | 71.54 | 71.00 | **71.59** |
| ResNet-34 | 73.93 | 73.78 | 74.36 | 73.15 | 73.93 |
| ResNet-18 | 70.16 | 71.44 | 71.33 | 71.36 | **71.62** |
| ResNet-101 | 76.62 | 76.80 | 78.22 | 76.28 | 77.81 |
| ResNet-50 | 75.99 | 76.70 | 76.70 | 76.78 | **77.10** |
| ResNet-101 | 77.85 | 78.17 | 78.36 | 78.00 | 78.31 |
| ResNet-18 | 70.16 | 70.80 | **71.09** | 70.70 | **71.07** |
| ResNet-18 | 70.16 | 70.80 | 71.08 | 70.63 | 70.94 |

Table 3: **Top-1 Accuracy (%) comparison on ImageNet with same architecture style teacher-student pairs.** The upper and lower models denote student and teacher, respectively.

| Backbone | Vanilla | DML | KDCL | SwitOKD | DML+ADM |
|---|---|---|---|---|---|
| 0.5MobileNetV2 | 62.49 | 64.21 | 64.32 | 63.94 | **65.10** |
| ResNet-18 | 70.43 | 69.00 | 70.75 | 68.81 | 69.66 |
| MobileNet-V1 | 68.81 | 71.13 | 70.76 | 71.07 | **71.69** |
| ResNet-50 | 76.38 | 75.44 | 76.42 | 74.80 | 75.68 |

Table 4: **Top-1 Accuracy (%) comparison on ImageNet with different architecture style teacher-student pairs.** The upper and lower models denote student and teacher, respectively.

### 5.3. Transfer Learning to Semantic Segmentation

Experiments are also conducted on semantic segmentation, which is recognized as a demanding dense prediction task. We train DeepLabV3 (Chen et al., 2018) and PSPNet (Zhao et al., 2017) with ResNet-18 backbone on Cityscapes dataset. Different pre-trained models are considered using an online teacher with ResNet-101 backbone. As the results summarized in Table 5, our ADM can significantly outperform existing OKD methods on semantic segmentation task.

| Methods | Teacher | Student | DML | KDCL | SwitOKD | DML+ADM |
|---|---|---|---|---|---|---|
| DeepLabV3 | 78.30 | 73.58 | 74.19 | 73.82 | 73.98 | **74.55** |
| PSPNet | 76.96 | 70.74 | 71.54 | 71.49 | 71.72 | **71.98** |

Table 5: **Results on Cityscapes val dataset.** All models are pretrained on ImageNet.

### 5.4. Extension to Diffusion Distillation

Figure 4 illustrates the dynamic consensus region, which is initially concentrated around the target object (t < 400), progressively diffusing to multi-scale regions (400 ≤ t < 600), and stabilizing globally (t ≥ 600). Simply, we employ the consensus loss when t < 400 during the training process, achieving promising results. We present the results for Inception Score (IS) (Salimans et al., 2016) and Fréchet Inception Distance (FID) (Heusel et al., 2017) for each method. Table 6 presents the results of step distillation experiments conducted on the CIFAR-10 dataset under the settings described in (Sun et al., 2023). When distilled to 1-step sampling, the FID score decreased by 2. Table 7 illustrates the model distillation experiments on the ImageNet dataset. We utilize the DiT-base(Peebles & Xie, 2023) model as the teacher to distill the DiT-small(Peebles & Xie, 2023) student model. We use MSE loss as our baseline. The FID score decreased by 1 and the IS increased by 3 after training for 400k steps.

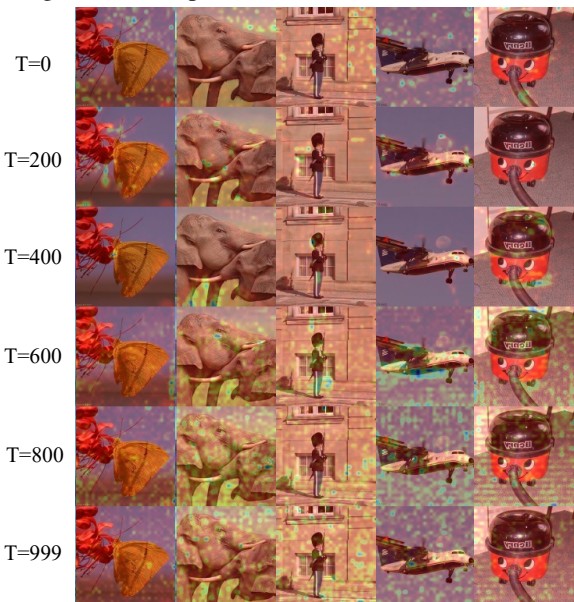

Figure 4: **Visualization of the spatial-temporal evolution with high teacher-student similarity.** Red indicates high similarity values. Best viewed in color.

### 5.5. Extension to Offline KD

ADM contains a teacher-discrepancy feature enhancement module and a student-similar feature knowledge distillation module. Evidently, the latter part is suitable for application

| Sampling Steps | Method | FID ($\downarrow$) | IS ($\uparrow$) |
|---|---|---|---|
| 1 | PD | 15.926 | 7.764 |
| | PD+ADM | **13.939** | **8.037** |
| 2 | PD | 7.322 | 8.653 |
| | PD+ADM | **6.830** | **8.760** |
| 4 | PD | 4.667 | 9.020 |
| | PD+ADM | **4.555** | **9.042** |

Table 6: **Results on Diffusion Step Distilltion.**

| Model/ Method | CFG=4.0 | 400K FID-50K | 400K IS | 300K FID-50K | 300K IS | 200K FID-50K | 200K IS | 100K FID-50K | 100K IS |
|---|---|---|---|---|---|---|---|---|---|
| DiT-B/2 | $\checkmark$ | 11.82 | 219.97 | 12.03 | 188.92 | 13.67 | 143.42 | 22.64 | 74.57 |
| (teacher vanilla) | | 42.78 | 33.50 | 47.50 | 29.43 | 55.29 | 24.61 | 72.21 | 17.70 |
| DiT-S/2 | $\checkmark$ | 14.83 | 123.75 | 17.05 | 105.53 | 21.94 | 79.20 | 36.13 | 43.62 |
| (student vanilla) | | 67.18 | 20.30 | 71.42 | 18.91 | 78.58 | 16.64 | 94.00 | 13.16 |
| DiT-S/2-MSE | $\checkmark$ | 13.67 | 136.73 | 15.75 | 112.74 | 18.93 | 93.63 | 31.98 | 48.85 |
| | | 62.08 | 22.27 | 65.75 | 20.77 | 72.71 | 18.30 | 86.43 | 14.52 |
| DiT-S/2-ADM | $\checkmark$ | **12.75** | **139.41** | **14.77** | **124.65** | **17.95** | **99.08** | **30.04** | **53.98** |
| | | **61.04** | **22.76** | **64.03** | **21.35** | **70.60** | **18.83** | **85.22** | **14.75** |

Table 7: **Results on Diffusion Model Distilltion.**

in offline KD. To further validate the ADM, we also compare ADM with offline knowledge distillation approaches including KD and DKD that require a fixed and pre-trained teacher. Specifically, we set $\alpha$, $\beta$, $\gamma$ in Eq.(9) and Eq.(10) as 1.0/0.0/0.0 and 0.1/0.0/0.0 for CIFAR-100 and ImageNet Dataset. Table 8 reveals that ADM achieves superior performance over offline fashions. For example, incorporating ADM based on KD resulted in a 1.33% improvement, and when applied to one of the state-of-the-art DKD methods, there was still an approximate increase of 0.3% on CIFAR-100 dataset. Table 9 shows that ADM increases KD accuracy by 0.22% and NORM 0.12% on ImageNet dataset.

## 5.6. Extension to Multiple Teachers and Students.

Similar to DML and SwitOKD, ADM can be extended for multiple networks easily. As shown in Table 11, We adopt the same three-network settings as SwitOKD. 1T2S means that one teacher model and two student models, and 2T1S means that two teacher models and one student model. We achieve a higher accuracy student across all methods in the 2T1S.

| Backbone | Vanilla | DML | SwitOKD (1T2S) | SwitOKD (2T1S) | DML+ADM (1T2S) | DML+ADM (2T1S) |
|---|---|---|---|---|---|---|
| MobileNet-V2(S) | 66.01 | 72.28 | 72.20 | 71.97 | 71.69 | **72.51** |
| WRN-16-2(S/T) | 72.79 | 74.78 | 74.91 | **75.07** | 74.80 | 74.09 |
| WRN-16-10(T) | 79.30 | 80.32 | 80.55 | 80.40 | 80.58 | **80.72** |

Table 11: **Accuracy (%) comparison with 3 networks on CIFAR-100.** WRN-16-2 serves as either teacher (T) or student (S) for DML, while is treated as S for 1T2S and T for 2T1S.

## 5.7. Ablation Studies

**Hyper-Parameters Tuning.** We explore the impact of three different hyperparameter values on the performance. Table

10 and 12 report the student and teacher accuracy with different $\alpha$, $\beta$ and different $\gamma$ in ImageNet. We use ResNet-34 and ResNet-18 as teacher and student models, respectively. Experiments demonstrate that the same hyperparameters can be employed for distillation across distinct model pairs, and the configuration of hyperparameters has little influence on the results. For $\gamma$, due to the magnitude of $L_{feat}$, it is generally set lower than 0.1. For $\alpha$ and $\beta$, the results indicate that both parameters demonstrate stable increases when set around 0.5. It is worth noting that we observe a higher accuracy when the weights of $\alpha$ and $\beta$ are set close to the mIoU of foreground objects, as obtained in Figure 2. This finding substantiates that ADM method is indeed learning in an anticipated manner.

**Effects of Feature Distillation Loss and ADM Loss.** This paper proposes three types of losses: $\mathcal{L}_{feat}$, $\mathcal{L}_{co}$ and $\mathcal{L}_{di}$. To validate the effectiveness of each loss, we conduct experiments to train students with these losses separately. The results on Table 13 verify that both $\mathcal{L}_{feat}$ and $\mathcal{L}_{adm}$ can outperform the DML, also, the performance could be further boosted by combining them together. It is noteworthy that $L_{di}$ loss effectively enhances the performance of the teacher model. We also investigate the individual effectiveness of ADM in Table 14.

| $\mathcal{L}_{feat}$ | $\mathcal{L}_{co}$ | $\mathcal{L}_{di}$ | stu_acc | tea_acc |
|---|---|---|---|---|
| | | | 71.12 | 73.78 |
| $\checkmark$ | | | 71.43 | 73.91 |
| | $\checkmark$ | | 71.35 | 73.78 |
| | | $\checkmark$ | 71.46 | 74.07 |
| | $\checkmark$ | $\checkmark$ | 71.50 | 73.91 |
| $\checkmark$ | $\checkmark$ | $\checkmark$ | 71.59 | 73.93 |

Table 13: **Effects of feature distillation loss and ADM loss.**

| Backbone | Vanilla | DML | ADM |
|---|---|---|---|
| S:ResNet-18 | 70.16 | 71.12 | **71.33** |
| T:ResNet-34 | 73.93 | 73.78 | 73.97 |
| S:0.5MobileNetV2 | 62.49 | 64.21 | **64.59** |
| T:ResNet-18 | 70.16 | 69.00 | **70.58** |

Table 14: **The individual effectiveness of ADM.** S: Student, T: Teacher.

**Different Types of ADM loss.** In this paper, we use cross-entropy(CE) to strengthen the ADM loss for brevity. Table 15 shows different forms of ADM loss. Adopting CE and KD loss helps us to achieve better performance.

| $\mathcal{L}_{co}$ | $\mathcal{L}_{di}$ | stu_acc | tea_acc |
|---|---|---|---|
| $\mathcal{L}_{ce}$ | $\mathcal{L}_{ce}$ | 71.41 | 74.03 |
| $\mathcal{L}_{kd}$ | $\mathcal{L}_{ce}$ | 71.59 | 73.93 |
| $\mathcal{L}_{kd}$ | $\mathcal{L}_{kd}$ | 71.45 | 73.98 |

Table 15: **Different types of ADM loss.**

**Combined with More Methods.** ADM is a plug-and-play module that can effectively enhance the performance

| Method | Same architecture style | | Different architecture style | | |
|---|---|---|---|---|---|
| | ResNet-32×4 | ResNet-56 | WRN-16-2 | ResNet-32×4 | ResNet-50 |
| | ResNet-8×4 | ResNet-20 | 0.5MobileNet-V2 | ShuffleNet-V2 | MobileNet-V2 |
| Teacher | 79.42 | 72.34 | 73.26 | 79.42 | 79.34 |
| Student | 72.50 | 69.06 | 64.60 | 71.82 | 64.60 |
| KD | 73.75 | 71.05 | 68.91 | 75.35 | 68.42 |
| KD+ADM | **74.52** | **71.46** | **70.24** | **75.73** | **69.04** |
| DKD | 76.24 | 71.52 | 69.03 | 77.07 | 70.35 |
| DKD+ADM | **76.66** | **71.58** | **69.68** | 77.03 | **70.64** |

Table 8: **Evaluation results in offline KD on CIFAR-100 dataset.** The upper and lower models denote teacher and student, respectively.

| | Teacher | Student | KD | KD+ADM | DKD | DKD+ADM | DisWOT | DisWOT+ADM | NORM | NORM+ADM |
|---|---|---|---|---|---|---|---|---|---|---|
| top-1 | 73.31 | 69.75 | 71.03 | **71.25** | 71.70 | **71.84** | 71.36 | **71.49** | 72.03 | **72.15** |
| top-5 | 91.42 | 89.07 | 90.05 | 90.12 | 90.41 | 90.36 | 90.23 | 90.23 | 90.41 | **90.54** |

Table 9: **Top-1 accuracy (%) in offline KD on the ImageNet validation.** We set ResNet-34 as the teacher and ResNet-18 as the student. All results are the average over 3 trials.

| $\alpha$ | 0.0 | 0.1 | 0.1 | 0.2 | 0.2 | 0.2 | 0.3 | 0.3 | 0.3 | 0.5 | 0.6 | 0.7 | 1.0 |
|---|---|---|---|---|---|---|---|---|---|---|---|---|---|
| $\beta$ | 0.0 | 0.5 | 0.7 | 0.5 | 0.6 | 0.8 | 0.5 | 0.6 | 0.7 | 0.5 | 0.2 | 0.3 | 1.0 |
| $stu\_acc$ | 71.12 | 71.42 | 71.37 | **71.48** | **71.59** | 71.48 | **71.69** | 71.60 | 71.37 | **71.52** | **71.55** | 71.43 | 71.31 |
| $tea\_acc$ | 73.78 | 73.92 | 73.90 | 73.88 | **73.93** | 73.77 | **74.11** | **74.13** | 74.07 | 74.05 | **74.20** | 74.12 | 74.10 |

Table 10: **Different $\alpha$ and $\beta$.**

| $\gamma$ | 0.01 | 0.1 |
|---|---|---|
| $stu\_acc$ | **71.59** | 71.48 |
| $tea\_acc$ | **73.93** | 73.91 |

Table 12: **Different $\gamma$.** $stu\_acc$ represents student's accuracy and $tea\_acc$ represents teacher's accuracy.

of existing methods. In this paper, when combined with DML, ADM can basically achieve sota in all settings, and when combined with stronger methods, better results can be achieved. As shown in Table 3, KDCL achieves the sota performance in some settings, and Table 16 shows several groups of comparative experiments combined with KDCL and SHAKE on the ImageNet dataset.

| Backbone | Vanilla | KDCL | KDCL+ADM | SHAKE | SHAKE+ADM |
|---|---|---|---|---|---|
| S:ResNet-18 | 70.16 | 71.54 | **71.78** (+0.24) | 71.65 | **71.75** (+0.10) |
| T:ResNet-34 | 73.93 | 74.36 | 74.34 | 73.45 | 73.50 |

Table 16: **Combined with KDCL.** S: Student, T: Teacher.

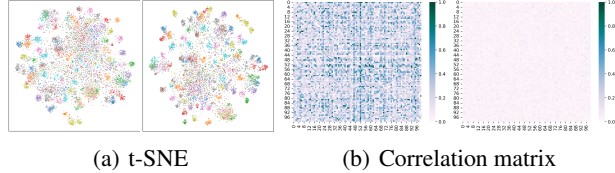

(a) t-SNE                    (b) Correlation matrix

Figure 5: **Visualzaiton**. Obviously, ADM (right) results in more discriminative features, reduced prediction discrepancies with teachers, and enhanced calibration than DML (left).

**Visualization.** In addition, we present visualizations from three perspectives in Figure 5 (with setting teacher as ResNet32×4 and student as ResNet8×4 on CIFAR-100).

## 6. Conclusion

In this paper, we present Asymmetric Decision-Making (ADM), an innovative online knowledge distillation method that unifies consensus and divergence feature learning and can seamlessly integrates with existing methods to leverage feature richness. Based on our insight into feature similarity, ADM employs a Consensus Learning on the student model to facilitate the learning of fundamental spatial knowledge, and a Divergence Learning on the teacher model to continually enhance its capabilities and drive the student model to learn and catch up. This dynamic interaction between the teacher and student models epitomizes the principle that "iron sharpens iron," thereby achieving superior performance in online knowledge distillation. Extensive experiments show our superiority in various benchmark tasks. We hope this paper will provide some insights for future OKD research.

## Acknowledgments

We gratefully acknowledge the enthusiastic members of the Fingerprint Algorithm Group at JIIOV Technology for their

helpful discussions.

## Impact Statement

This work advances the field of online knowledge distillation through unifying consensus and divergence feature learning. The technique may benefit the deployment of lightweight models on resource-constrained devices.

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
