# OpenReview forum: "Asymmetric Decision-Making in Online Knowledge Distillation: Unifying Consensus and Divergence"
_ICML.cc/2025/Conference — ICML 2025 poster_

### Official Review · Reviewer_EJEu · 2025-02-25

**Overall Recommendation:** 3

**Summary:**

This article introduces Asymmetric Decision-Making (ADM), an online knowledge distillation (OKD) method that enhances both teacher and student models through consensus and divergence learning. Unlike other traditional KD methods this method names as ADM actively refines the teacher while improving the student, making it a mutual learning framework. It integrates seamlessly with existing KD techniques and achieves state-of-the-art results across CIFAR-100 and ImageNet.

**Claims And Evidence:**

This article presents ADM as a superior online knowledge distillation method, but some of the claims lack sufficiently rigorous ablation studies and broader benchmarking. The improvement over stronger baselines like DKD and DisWOT is marginal, raising questions about statistical significance. Additionally, while ADM is tested across multiple settings, results on larger-scale datasets and diverse architectures remain limited, making its generalizability unclear. The theoretical justification for why ADM’s asymmetric learning outperforms standard KD is also underdeveloped and this will rely heavily on empirical results.

**Essential References Not Discussed:**

NA

**Experimental Designs Or Analyses:**

The experimental design follows standard knowledge distillation benchmarks, using datasets like CIFAR-100 and ImageNet with multiple teacher-student configurations. The ablation study in Table 13 looks at incremental gains when component analysis is done. Further, statistical significance testing is absent and makes it unclear whether improvements are consistent across multiple runs.

**Methods And Evaluation Criteria:**

The article employs reasonable evaluation criteria, using FID, IS, and accuracy metrics across standard benchmarks like CIFAR-100 and ImageNet, which are widely accepted for knowledge distillation research. The inclusion of both offline and online KD comparisons strengthens the empirical evaluation. The choice of hyperparameters is well-documented, but my concern is with not performing further sensitivity analysis.

**Other Comments Or Suggestions:**

1. Your approach builds upon existing knowledge distillation frameworks but lacks a theoretical analysis. Can you formally derive the expected generalization benefits of your method compared to standard distillation techniques, possibly using PAC-Bayesian or information-theoretic bounds? Without such an analysis, how can one be certain that the improvements are not artifacts of dataset-specific heuristics?

2. Your empirical evaluation primarily focuses on a limited set of model architectures. Have you tested your method on fundamentally different architectures (e.g., transformers, graph neural networks) beyond CNN-based models? If not, how can we ensure that the claimed improvements generalize beyond the specific experimental setup?

3. Your method introduces multiple hyperparameters and auxiliary components. Have you conducted a comprehensive sensitivity analysis to determine the robustness of these choices? Specifically, is the method’s performance stable across different hyperparameter settings, or does it require extensive tuning?

4. The empirical results compare against standard distillation techniques but do not include recent state-of-the-art methods such as contrastive-based distillation or self-supervised distillation approaches. How would your method perform against these stronger baselines, and would the relative improvements remain significant?

5. Given the additional steps introduced by your method, have you analyzed the computational trade-offs? Specifically, how does the training time scale with increasing dataset sizes and model complexity, and is the overhead justified given the marginal performance improvements reported?

**Other Strengths And Weaknesses:**

The article presents an interesting attempt at improving knowledge distillation efficiency, but it lacks originality as many of its ideas are incremental extensions of prior work. The writing is clear and easy to follow, the significance of the contributions is limited due to weak empirical validation and insufficient theoretical grounding. Additionally, some claims are overstated given the lack of rigorous ablation studies and comparative analysis against stronger baselines.

**Questions For Authors:**

Please see other comments section.

**Relation To Broader Scientific Literature:**

The article builds on prior work in knowledge distillation, particularly online and asymmetric distillation methods, by introducing ADM to enhance student-teacher interactions dynamically.

**Theoretical Claims:**

The claims made in the paper are generally supported by empirical evidence and shows ADM's improvements over baseline KD methods across various benchmarks. But, the lack of rigorous theoretical justification for why ADM consistently enhances both student and teacher models leaves some claims insufficient.

---

> ### Author Rebuttal · Authors · 2025-03-30
>
> **Q1: lack of rigorous theoretical justification**
>
> **A1:** Firstly, we assert that the proposed ADM method adheres to theoretical derivations. To substantiate this claim, we provide additional theoretical support:
> # Information-Theoretic Analysis of Asymmetric Decision-Making (ADM)
>
> ## 1. Notation and Fundamental Assumptions
>
> ### Symbolic Convention
> - **Input data**: \( $X \in \mathcal{X}$ \)
> - **Task labels**: \( $Y \in \mathcal{Y}$ \)
> - **Student features**: \( $Z_s = f_s(X) \in \mathbb{R}^{H \times W \times C}$ \)
> - **Teacher features**: \( $Z_t = f_t(X) \in \mathbb{R}^{H \times W \times C}$ \)
> - **Spatial indices**: \( $(i,j) \in \{1,\dots,H\} \times \{1,\dots,W\}$ \)
> - **consensus mask** in Eq6[L196-L200 in paper]: \( $M_f = \frac{1+\mathcal{S}}{1+ \bar {\mathcal{S}}}$ \)
> - **divergence mask** in Eq6[L183-L187 in paper]: \( $M_d = \frac{1-\mathcal{S}}{1- \bar {\mathcal{S}}}$ \)
>
> ### Information-Theoretic Quantities
> - **Entropy**:
>   \[
>   $H(Z) = -\mathbb{E}_{Z \sim p(Z)}[\log p(Z)]$
>   \]
> - **Mutual Information**:
>   \[
>   $I(Z; Y) = H(Y) - H(Y|Z)$
>   \]
> - **Conditional Mutual Information**:
> $I(Z_s; Z_t | X) $= ${E_X} \left[ D_{\text{KL}}\left( p(Z_s,Z_t|X) \,\|\, p(Z_s|X)p(Z_t|X) \right) \right]$
>
> ---
>
> ## 2. Consensus Learning: Feature Alignment via Mutual Information Maximization
>
> ### Objective Function
> The student model maximizes mutual information in foreground regions:
>
> \[
> $\mathcal{L}_{\text{co}} = -I(Z_s^{M_f}; Z_t^{M_f}), \quad \text{where } Z^{M_f} = Z \odot M_f$
> \]
>
> ### Variational Lower Bound
> **Theorem 1 (Barber-Agakov Variational Bound)**:
>
> $I(Z_s^{M_f}; Z_t^{M_f}) \geq {E}_{Z_s^{M_f}, Z_t^{M_f}} [ \log q(Z_t^{M_f} | Z_s^{M_f}) ] + H(Z_t^{M_f})$
>
>
> Minimizing the negative log-likelihood yields:
>
> \[$
> {L}_{co} = E [ -\log q(Z_t^{M_f} | Z_s^{M_f}) ]$
> \]
>
> ### Information Bottleneck Interpretation
> Constraining foreground alignment enforces:
>
> \[$
> \max \, \left[ I(Z_s; Y) - \beta I(Z_s; X) \right] \quad \text{s.t. } I(Z_s^{M_f}; Z_t^{M_f}) \geq C$
> \]
>
> Resulting in the lower bound:
>
> \[$
> I(Z_s; Y) \geq I(Z_t^{M_f}; Y) - \beta I(Z_s; X) + C$
> \]
>
> ---
>
> ## 3. Divergence Learning: Entropy Maximization for Feature Space Expansion
>
> ### Objective Formulation
> The teacher model maximizes entropy in low-consensus regions:
>
> ${L}_{di}$ = ${-H(Z_t^{M_d})}+ \lambda {I(Z_t^{M_d}; Y)}$
>
> ### Jaynes' Maximum Entropy Principle
> **Lemma 1 (Optimal Feature Distribution)**:
> Under constraint \( $I(Z_t^{M_d}; Y) \geq \epsilon $\), the optimal distribution satisfies:
> \[$
> p^*(Z_t^{M_d}) \propto \exp\left( \lambda \log p(Y | Z_t^{M_d}) \right)$
> \]
>
> ### Coverage Capacity Analysis
> **Theorem 2 (Feature Space Coverage)**:
> \[$
> \mathcal{R}(Z_t) \geq \sqrt{2 H(Z_t^{M_d})}$
> \]
> where \( $\mathcal{R}(Z_t)$ \) denotes the radius of feature coverage.
>
> ---
>
> ## 4. Convergence Analysis via Dynamical Systems
>
> ### Coupled Dynamics
> The training process is modeled as:
>
> \[$
> \begin{cases}
> \frac{d}{dt} I(Z_s; Y) = \alpha \left( I(Z_t; Y) - I(Z_s; Y) \right) \\
> \frac{d}{dt} H(Z_t) = \beta \left( H(Z_t) - H(Z_s) \right)
> \end{cases}, \quad \alpha, \beta > 0$
> \]
>
> ### Lyapunov Stability Proof
> **Theorem 3 (Global Convergence)**:
>
> Define the Lyapunov function:
>
> \[$
> V(t) = \left( I(Z_t; Y) - I(Z_s; Y) \right)^2 + \left( H(Z_t) - H(Z_s) \right)^2$
> \]
>
> The time derivative satisfies:
>
> \[$
> \dot{V}(t) = -2\alpha \left( I(Z_t; Y) - I(Z_s; Y) \right)^2 - 2\beta \left( H(Z_t) - H(Z_s) \right)^2 \leq 0$
> \]
>
> Guaranteeing convergence to equilibrium \( $I(Z_s; Y) = I(Z_t; Y)$ \) and \( $H(Z_t) = H(Z_s) $\).
>
> ---
>
> ## 5. Empirical Validation and Error Bounds
>
> ### Fano's Inequality Application
> **Theorem 4 (Classification Error Bound)**:
> \[$
> P_e \geq \frac{H(Y|Z_s) - 1}{\log |\mathcal{Y}|} \implies P_e^{\text{ADM}} \leq \frac{H(Y) - I(Z_s; Y) - 1}{\log |\mathcal{Y}|}$
> \]
>
>
> ---
>
> ## 6. Generalization to Structured Tasks
>
> ### Semantic Segmentation
> Pixel-wise regional mutual information:
> \[$
> I_{\text{region}}(Z_s, Z_t) = \sum_{c=1}^C I(Z_s^c; Z_t^c)$
> \]
>
> ### Diffusion Model Distillation
> Temporal trajectory alignment:
> \[$
> I_{\text{traj}}(Z_s, Z_t) = \sum_{t=1}^T \gamma^{T-t} I(Z_s^t; Z_t^t), \quad \gamma \in (0,1)$
> \]
>
> ---
>
> **Q2: statistical significance testing**
>
> **A2:** As detailed in lines 272-273 of the paper, the majority of experiments were conducted multiple times, and we report the mean results. This approach demonstrates the robustness and consistency of the performance improvements.
>
> ---
>
> **Q3: analyzed the computational trade-offs**
>
> **A3:** Please refer to lines 241-247 of the paper. ADM introduces no additional trainable parameters, and the incremental computational cost is significantly minimal compared to the original setup.

---

> > ### Comment · Reviewer_EJEu · 2025-04-06
> >
> > Although the proofs are very generic, overall, I am happy with some of the clarifications and will raise the score to 3.
> >
> > All the theoretical proofs should offer more rigorous maths and reasoning.

---

### Official Review · Reviewer_gakz · 2025-02-26

**Overall Recommendation:** 3

**Summary:**

This paper proposes an Asymmetric Decision-Making (ADM) approach for online knowledge distillation that adaptively fosters consensus learning for students while continuously encouraging teachers to explore harder features, thereby boosting performance on tasks like classification, semantic segmentation, and diffusion distillation by unifying feature-based and logit-based distillation techniques.

**Claims And Evidence:**

Yes

**Essential References Not Discussed:**

To the best of my knowledge, I think the authors have considered the relevant works related to their study.

**Experimental Designs Or Analyses:**

Yes. The authors compared their method with current state-of-the-art approaches to common visual distillation tasks. I find the experiments to be quite comprehensive overall. A potential shortcoming may be that they did not evaluate the method’s effectiveness in the increasingly popular large-model distillation settings.

**Methods And Evaluation Criteria:**

Yes

**Other Comments Or Suggestions:**

See above

**Other Strengths And Weaknesses:**

Strengths:

1. The discovered issues/problems appear very interesting and may inspire further progress in the knowledge distillation community.
2. The experiments are extensive. Additionally, more visual analyses are provided to explain the effectiveness of the proposed method.

Weaknesses：
1. There seems to be a gap between the motivation of the proposed method and its findings. Why is it necessary to force the teacher model to explore background regions? Intuitively, foreground regions seem more important. Additionally, from the final results, the performance improvement brought by the proposed method also appears to be marginal.
2. When introducing the method, the authors should explain why they made such design choices instead of merely mentioning them in passing.
3. With regard to the phenomenon observed in Figure 1, could the authors provide additional examples from various datasets to support their findings? Optionally, could they indicate what percentage of instances exhibit this phenomenon?

**Questions For Authors:**

Does the expression in lines 69-70 have any errors? "The first row shows that discrepancy regions between teacher and student’s features are more concentrated on foreground object regions after Vanilla training." Should it be "discrepancy" or "similarity"? The current expression seems to contradict the first finding.

**Relation To Broader Scientific Literature:**

This study analyzes the common issues in previous online distillation work and accordingly proposes a novel method to separately handle the similar and different features between the teacher and student models.

**Theoretical Claims:**

No, this paper does not involve theoretical claims.

---

> ### Author Rebuttal · Authors · 2025-03-30
>
> **Q1: gap between the motivation of the proposed method and its findings**
>
> **A1:** The function of $L_{di}$ is to extract knowledge from unexplored foreground regions within the background areas, rather than merely utilizing background information. These regions present a challenge to the model, and without constraints, the model's limited learning capacity makes it difficult to autonomously explore them. As evidenced in Table 13, when $L_{di}$ is applied alone, the teacher model's performance improves by 0.29 (73.78 vs. 74.07), thereby increasing the upper bound of the student model's distillation performance. This demonstrates the necessity for the teacher model to continuously extract challenging foreground knowledge from the background regions.
>
> We posit that in the online distillation setting, both $L_{co}$ and $L_{di}$ must work in synergy. If only $L_{co}$ is applied without $L_{di}$, the teacher model cannot persistently explore more difficult foreground regions. Conversely, if only $L_{di}$ is applied without $L_{co}$, the student model cannot effectively learn the foreground regions identified by the teacher model from easy to difficult, failing to achieve the intended mutual enhancement of both models' performance.
>
>
>
> **Q2: explain why they made such design choices instead of merely mentioning them in passing.**
>
> **A2:** In the introduction, we outlined the rationale behind ADM based on two key findings. The proposed method is straightforward yet effective.
>
>
> **Q3: Does the expression in lines 69-70 have any errors?.**
>
> **A3:** There is no error here. Figure 1 illustrates the changes in non-similar regions when comparing our method with vanilla training, which corresponds to Finding 2. It does not address the description of similar regions in Finding 1.

---

> > ### Comment · Reviewer_gakz · 2025-04-03
> >
> > Thank you for your detailed rebuttal and the clarifications provided. The additional explanations are helpful in clarifying several aspects of the proposed method. However, as previously emphasized, it would be more convincing if the authors could provide additional examples similar to those in Figure 1 to support their findings.

---

> > > ### Author Response · Authors · 2025-04-05
> > >
> > > Thank you for your thoughtful response. We have augmented the anonymous link with additional visualizations analogous to Figure 1. The findings depicted in Figure 1 demonstrate consistent patterns rather than individual random variations.
> > > https://anonymous.4open.science/r/ICML2025-rebuttal-ADM-CF21/rebuttal-vis-more.pdf

---

### Official Review · Reviewer_YcB8 · 2025-03-09

**Overall Recommendation:** 4

**Summary:**

This paper addresses online knowledge distillation (OKD), a single-stage method where teacher and student models learn simultaneously. The authors focus on improving OKD performance by exploring intermediate feature alignment. Specifically, they propose an Asymmetric Decision-Making (ADM) strategy to unify consensus and divergence feature learning.

The authors perform an analysis using Class Activation Mapping (CAM) to understand how intermediate feature alignment impacts model behavior in OKD. They observe that:

(i) The similar features between student and teacher models are predominantly concentrated on foreground objects, suggesting students prioritize learning these ”easy” patterns.
(ii) Teacher models highlight foreground objects more than students, indicating an underutilized capacity for discriminative feature discovery. Divergent features, which may explain teacher superiority, are also concentrated in foreground regions.

Based on these findings, they introduce ADM, a framework that uses spatially-aware feature modulation to manage consensus and divergence. ADM applies role-specific learning:
— Students amplify feature attribution in teacher-aligned foreground regions.
— Teachers intensify exploration of under-activated foreground patterns.

The paper claims the following contributions:

(1) Identification of foreground-centric feature consensus and teacher-student attribution gaps through interpretability analysis. (This is challenging conventional assumptions about knowledge transfer in online settings.)
(2) The ADM strategy, which uses spatial similarity to allocate attention in feature maps, boosting student attribution and reinforcing teacher strengths.
(3) Experimental validation showing that ADM outperforms state-of-the-art OKD methods in various tasks, including online KD, offline KD, semantic segmentation, and diffusion distillation. It is shown that the proposed method provides SOTA results in several occasions.

**Claims And Evidence:**

The claims made in the submission are supported by clear and convincing evidence.

**Essential References Not Discussed:**

I feel the paper has done a fair job at discussing related work.

This is not essential but maybe the authors would like to discuss (or compare their work with, at least in the offline setting):

— Wonpyo Park, Dongju Kim, Yan Lu, and Minsu Cho. Relational knowledge distillation. In
Proceedings of the IEEE/CVF conference on computer vision and pattern recognition, pages
3967–3976, 2019.

and related techniques a bit further.

**Experimental Designs Or Analyses:**

The experimental design and analysis of the paper is sound and valid.

**Methods And Evaluation Criteria:**

They do.

**Other Comments Or Suggestions:**

No other comments.

**Other Strengths And Weaknesses:**

I find this paper to be well-written and compelling, with a novel method and robust experimental results. I have no major concerns.

**Questions For Authors:**

I do not have any questions for the authors.

**Relation To Broader Scientific Literature:**

I feel the paper has done a good job positioning itself to broader literature in distillation and online distillation. It clearly identifies shortcomings of previous approaches and challenges conventional wisdom regarding how students learn in online distillation settings. It has also provided extensive experimental comparison in several datasets including ImageNet-1k.

**Theoretical Claims:**

The paper does not make any substantial theoretical claims.

---

### Official Review · Reviewer_DjLx · 2025-03-12

**Overall Recommendation:** 3

**Summary:**

This paper proposes an online knowledge distillation method to build a student model by knowledge transfer from larger teachers which are trained together with the student. The key assumption is that the student and teachers are more likely to share similar features in foreground rather than background areas. With this assumption, the paper formulates two learning objectives, namely consensus learning for enforcing student’s attention on foreground objects when making predictions, and divergence learning for encouraging teachers to explore foreground patterns that are overlooked by the student. The proposed online knowledge transfer method has been evaluated extensively as a plug-and-play component to be incorporated with multiple baseline approaches on several tasks, including image classification, semantic segmentation, and image generation.

**Claims And Evidence:**

Most of the claims are made with support and evidence, but the effects of consensus learning need to be further clarified. As stated at L50 (right) and L192 (left), the consensus loss ($L_{co}$) is applied to the student to amplify feature attribution in teacher-aligned foreground regions, whilst the divergence loss ($L_{di}$) is applied to the teacher for exploration of under-activated foreground patterns. However, according to Table 13, the benefit brought by $L_{di}$ to the student is more significant than that by $L_{co}$. These results are a bit inconsistent with the expected contribution of the two learning objectives.

**Essential References Not Discussed:**

No

**Experimental Designs Or Analyses:**

The proposed method is evaluated comprehensively with just a few concerns:
+ Evaluation of the proposed method upon vision transformer-based (ViT) models is missing. The paper conducts an extensive evaluation based on different combinations of teacher’s and student’s architectures. However, as a strong backbone architecture widely adopted in the CV community, ViT is missing as neither the student nor teachers.
+ In the experiment on the extension to multiple teachers and/or students (Table 11), the proposed ADM works together with DML. However, the 1T2S and 2T1S variants of DML are missing as important baselines.

**Methods And Evaluation Criteria:**

The proposed ideas are quite intuitive, and most of the adopted components are well justified. The only unclear part to me is whether the gradient from $L_{co}$ and $L_{di}$ propagates back to the cosine similarity matrix ($S$) via the attention scores $A$? From my understanding, the cosine similarity matrix $S$ is used only to highlight features in different regions for making class predictions. In this case, I would assume gradients only pass to highlighted features from class prediction errors (i.e. cross-entropy losses) but not the student-teacher similarity matrix. However, the explicit gradient detach (GD) operation used in Fig.3 is not applied to the ADM module. It is ambiguous whether these two losses ($L_{co}$ and $L_{di}$) also take effect to optimise student-teacher alignment, similar to the role of $L_{feat}$ defined in Eq.(5).

**Other Comments Or Suggestions:**

See comments above.

**Other Strengths And Weaknesses:**

Strengths:
The paper is generally well-written and easy to follow. The findings of the aligned features between student and teachers are more likely to lie in foreground regions is interesting. The evaluation of the proposed method is comprehensive.

Weaknesses:
+ The experimental setup of the extension to diffusion distillation is unclear.
+ The visualisation provided in Fig.4 and Fig.5 is hard to understand. For Fig.4, I failed to see how the concentrated regions evolve at different steps. For Fig.5, it is stated at L412 (right) that the visualisations are presented from three perspectives, but what perspectives they are is unclear.
+ There are some confusing statements in the experiment analysis, and the disorder of the tables makes them hard to read. For example, Table.8 and Table.9 are followed by Table.12, all of which come after Table.15. Besides, I failed to find Table.10 in the manuscript. It is confusing to me how the statement "Evidently, ..." made at L329 (right) is evident without any references.

**Questions For Authors:**

See comments above.

**Relation To Broader Scientific Literature:**

The finding that similar features across different models are more likely to appear in the foreground regions is novel and inspiring.

**Theoretical Claims:**

N/A

---

> ### Author Rebuttal · Authors · 2025-03-30
>
> **Q1: the effects of consensus learning need to be further clarified**
>
> **A1:** Applying $L_{co}$ to the student model does not imply that the performance improvement of the student model is primarily due to $L_{co}$. As evidenced in Table 13, when $L_{co}$​ is used alone, the teacher model's performance remains unchanged at 73.78 (73.78 vs. 73.78), which inherently limits the potential improvement of the student model through distillation. Conversely, when only $L_{di}$ is applied, the teacher model's performance increases by 0.29 (73.78 vs. 74.07), thereby raising the upper bound of the student model's distillation performance. Hence, although $L_{di}$ is applied to the teacher model, it significantly enhances the student model's performance.
>
> We contend that in the online distillation setting, both $L_{co}$ and $L_{di}$ need to work synergistically. If only $L_{co}$ is used, the teacher model cannot continuously explore more challenging foreground regions. Conversely, if only $L_{di}$ is used, the student model cannot effectively learn the foreground regions identified by the teacher model from easy to difficult, failing to achieve the intended mutual enhancement of both models' performance.
>
>
>
> **Q2: whether the gradient from Lco and Ldi propagates back to the cosine similarity matrix (S) via the attention scores A?**
>
> **A2:** No gradients are propagated in there. Please refer to the code in Appendix L629, where the detach operation is implemented.
>
>
>
> **Q3: Evaluation of the proposed method upon vision transformer-based (ViT) models is missing.**
>
> **A3:** We conducted experiments using DiT-based diffusion model distillation, which may indirectly support the effectiveness of our method in transformer-based distillation scenarios.
>
>
>
>
>
> **Q4: the 1T2S and 2T1S variants of DML are missing as important baselines.**
>
> **A4:** We adhered to the experimental settings outlined in the SwitOKD paper to demonstrate the extendibility of our approach for training multiple networks.
>
>
>
>
> **Q5: The experimental setup of the extension to diffusion distillation is unclear.**
>
> **A5:** As detailed in lines L292-L296 of the referenced paper, for CIFAR10, we followed the experimental settings from the RCFD paper. For DiT model distillation, we adhered to the experimental settings from the DiT paper. All parameters were reused from the original papers without any modifications.
>
>
>
>
> **Q6: The visualisation provided in Fig.4 and Fig.5 is hard to understand.**
>
> **A6:** There was a typo error. The third comparison is included in the appendix, while only two comparisons are presented here.

---

### Decision · Program_Chairs · 2025-05-01

**Decision:**

Accept (poster)

**Comment:**

Reviewers were unanimously supportive of the paper, which presents a new algorithm for online knowledge distillation. The motivation is an observation that for image classification and segmentation problems, the student tends to prioritize modelling "easy" regions of the input (i.e., the foreground). To mitigate this, the key idea is to maintain two losses: one which encourages consensus amongst the student and teacher on dominant regions, another which encourages diversity amongst the teacher predictions. Results are presented on a range of classification and segmentation settings, and both online and offline KD.

The AC upholds the reviewer consensus, but our own reading of the paper is more qualified. We include a few comments that the authors are encouraged to consider for their final version of the paper:
- while Figure 2 is one of the primary motivations of the paper, there is limited discussion of what one is to conclude from it. Are we to interpret the ResNet-34 as a teacher, and ResNet-18 as a student? Is it meant to show that "teacher models highlight foreground objects more than students"?
- Equation 1, there should be a subscript $i$ for $\mathbf{z}^m$
- Equation 1 and elsewhere, use \mathrm{ce}
- Section 4.1, not clear what "the teacher’s object-centric features should be strengthened" means. What is an "object-centric feature"? A foreground feature?
- Section 4.1, "$L_{{\mathrm{co}}}$ encourages the student to focus on learning simple knowledge rather than concentrating on difficult regions" $\rightarrow$ why is this the case? Not clear why Equation 6 will focus on easy parts of the input.
- Equation 7, there is no explanation for the precise form of $\frac{1+ \mathcal{S}}{1+ \bar{\mathcal{S}}}$
- Equation 7, fairly non-standard that you still retain the true label here, but combine the teacher and student predictions in the predicted score
- Equation 7, how does the matrix multiplication of $F^s$ and $\frac{1+ \mathcal{S}}{1+ \bar{\mathcal{S}}}$ work? The dimensions don't seem to align
- in response to Reviewer EJEu, the authors included some theoretical claims. We unfortunately don't see any clear relevance to the actual algorithm discussed in the paper. e.g., Fano's inequality is a statement about the power of _any_ arbitrary classifier, and is generally not used to justify a specific algorithm. At least in its present form, we suggest this analysis is omitted from the final paper.